# Those who tan and those who don't: A natural experiment on colorism

**Tamar Kricheli Katz**[1☉]*, **Tali Regev**[2☉], **Shay Lavie**[1], **Haggai Porat**[3], **Ronen Avraham**[1]

**1** Buchmann Faculty of Law, Tel Aviv University, Tel Aviv, Israel, **2** The Interdisciplinary Center Herzelyia, Tel Aviv, Israel, **3** Harvard University, Cambridge, Massachusetts, United States of America

☉ These authors contributed equally to this work.
* tamarkk@taux.tau.ac.il

**Data Availability Statement:** The NLSY (97) is publicly available. Researchers who wish to access the Geocodes data should apply directly to the NLSY. Data requests are available at www.bls.gov/nls/geocodeapp.htm.

## Abstract

Are darker-skinned workers discriminated against in the labor market? Studies using survey data have shown that darker skin tone is associated with increased labor market disadvantages. However, it is hard to refute the possibility that other factors correlated with skin tones might affect employment outcomes. To overcome this inherent limitation, we use a natural experiment: we utilize changes in one's own skin tone, generated by exposure to the sun, to explore the effect of skin tone on the tendency to be employed. We find that those people whose skin tone becomes darker by exposure to the sun (but not others) are less likely to be employed when the UV radiation in the previous three weeks in the area in which they reside is greater. These within-person findings hold even when controlling for the week, the year, the region, demographic characteristics and the occupation and industry one is employed in.

## Introduction

Studies have documented persistent disparities in earnings and other employment outcomes between blacks and whites in the US [1–4]. Experimental evidence further indicate that employers are reluctant to hire black workers [3,5] and that black men are viewed as unreliable, incompetent, threatening, and less motived and committed than white men [6–11]. Recently, several studies on colorism have provided evidence that intraracial differences in skin tone also matter for labor force and other stratification outcomes [12–18]: Even within-race, people with darker skin tones tend to be employed less and to earn less compared to people with lighter skin tones. These studies have documented greater effects of skin tones for men compared to women [19].

Although survey data provide an opportunity to observe patterns of employment outcomes by race (or by skin tone) in the entire labor force, it is difficult to use them to document causality: it is nearly impossible to rule out the possibility that unmeasured differences between people (like pre-labor market discrimination or differences in the quality of education) generate the observed differences in employment outcomes. Previous studies have dealt with this issue of causality by taking into account measures of academic achievement and cognitive skills [20] and comparing the employment outcomes of black and white employees with similar such

**Funding:** The authors received no specific funding for this work.

**Competing interests:** The authors have declared that no competing interests exist.

measures. However, this approach might fail to fully hold constant all the unobserved relevant traits of individuals.

Another prominent methodological approach to deal with the issue of causality is to use field experiments in which job applications of fictitious job applicants who vary by whether they have white- or black- sounding first and last names are sent to real employers [5]. However, field experiments tend to focus on a small number of occupations and industries (those that involve formal applications) and therefore fail to account for discrimination in the entire labor force. In addition, because of methodological constrains, field experiments tend to use callbacks for interviews (and not the actual hiring) as outcomes. Finally, black-sounding first and last names tend to be associated with lower classes and therefore it is hard to disentangle the effects of class and race.

We take a different approach to deal with the inherent limitation of inferring causality using survey data. We use a natural experiment to explore the effects of skin tone on employment; we follow people over time and test for the effects of changes in their skin tone—generated by exposure to the sun—on their tendency to be employed. We therefore build on the literature that shows that darker skin tones are associated with labor force disadvantages, and ask whether the same is true even within-person so that the same individual experiences disadvantages when she looks darker.

We use the UV radiation in one's metropolitan area in the preceding three weeks as an exogenous variable. Whereas UV radiation can darken one's complexion, it does not influence one's skills, commitment or other related characteristics that might affect one's employability. Thus, differences in the tendency to be employed when UV radiation is greater cannot be attributed to changes in one's employment related characteristics. Moreover, exposure to UV radiation affects people with different skin tones differently: some people—those with medium, moderate brown, and dark brown skin—look darker when exposed to the sun. People with moderate brown skin for example may look like they have a dark brown skin-tone after tanning. People with white and pale white complexion look different after spending time in the sun, however they do not look darker, but burnt. People with very dark brown to dark skin do not look different after spending time in the sun. If indeed people discriminate on the basis of skin tone, we would expect the people who are prone to tanning (those with medium, moderate brown, and dark brown skin tones) to be discriminated against more when UV radiation is greater and tanning occurs. However, we would not expect people who cannot tan (those with white and pale white skin tones) or people for whom the change in their skin color likely goes unnoticed by a casual observer (very dark brown to dark skin tones) to be affected by being exposed to greater UV radiation.

Using the 1997 National Longitudinal Survey of Youth (NLSY97), we find that indeed, those people whose skin tone becomes darker by exposure to the sun (but not all others) are less likely to be employed when exposed to greater UV radiation. These within-person findings hold even when controlling for the week, the year, the region, demographic characteristics and the occupation and industry one is employed in. A separate analysis for women and men reveals that it is the effect of UV radiation on men's employment, but not on women's, that drives the results we present. We further show that our results are robust to various specifications.

Note that our focus here is on discrimination on the basis of one's skin tone. Although we use the tendency of individuals to look darker when exposed to the sun as a variable, we do so for methodological reasons; tanning changes the perceived skin tones of individuals. Because race and skin tones are immediately noticed and encoded, one's perceived skin tone is an important factor in the attempt to empirically identify discrimination [21]. In other words, we care less about discrimination on the basis of tanning and much more about discrimination

on the basis of one's skin tone. Moreover, we do not aim to capture the full range of experiences associated with race and racism, but rather to provide evidence for labor force discrimination on the basis of one's skin tone.

Finally, with the methodological approach we use, we only expose the tip of the iceberg: we only estimate the effects of *changes* in one's perceived skin tone on her employment status. We do not estimate the direct effects of skin tone on employment outcomes nor do we estimate the effects of one's skin tone in other arenas of life.

## Data and methods

The NLSY97 is a longitudinal dataset that follows the lives of a sample of several thousand Americans who were born between 1980–84. It provides rich information on personal characteristics, including the employment status for each week through the surveyed year. The NLSY97 interviews respondents only once every year. This means that the weekly employment status is based on respondents' recollections and may therefore not be completely accurate. Statistically significant effects therefore might be harder to find.

The NLSY97 also includes data on respondents' skin tone that was collected once in 2008. The date in which the skin tone was recorded by the interviewer is also available. In Fig 1 we present the NLSY skin color rating card used by interviewers.

We matched the NLSY97 with data on the average weekly UV radiation in each respondent's area of residence. We drew the UV data from a dataset collected by the National Oceanic and Atmospheric Administration (NOAA), which gauges UV radiation in 58 metropolitans in the US according to the physical location of the 58 stations. Our sample is therefore restricted to those respondents residing in metropolitan areas where a weather station exists, leaving out respondents residing elsewhere. The NOAA uses the common index of ground UV (largely UV-A) radiation, which is primarily related to the elevation of the sun in the sky, the amount of ozone in the stratosphere, and the amount of clouds present. Hence, UV levels are typically greater in the summer. The UV index ranges from 0 (lowest) to 15–16 (highest), where higher levels of UV render a greater effect on one's skin.

Thus, for each of the respondents we have weekly data on her employment status as well as the average UV radiation in the area in which she resides, together with other demographic and employment characteristics (such as her occupation and industry). We use person fixed effects models to predict the effects of the average UV radiation in the previous three weeks on the employment status of respondents in the current week.

The effects of UV radiation on people's skin tone vary by skin type and initial skin tone. In general, UV radiation tends to generate a darker skin tone for those with medium, moderate brown, and dark brown skin, but not for those with white, pale white, and very dark brown to dark skin. Those with white or pale skin tend to burn but not tan. Those with very dark brown to dark skin typically absorb UV radiation, with only minimal change to their complexion.

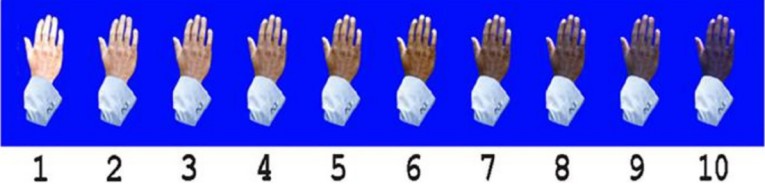

**Fig 1. NLSY skin color rating card.**

The Fitzpatrick scale that was developed by Thomas B. Fitzpatrick describes the response of different skin types to UV radiation [22, 23] (see Fig 1A in the Appendix for the Fitzpatrick scale). Building on the Fitzpatrick scale, we grouped the NLSY skin color ratings into three skin tone categories: two categories of people who do not tend to tan—lightest skin tones (skin tones 0–1) and darkest skin tones (skin tones 7–10)—and one category of people who do tend to tan—intermediate skin tones (skin tones 2–6) [22, 23]. We later run some robustness checks where we group the skin tones differently.

The time required for tanning to fully manifest following exposure to UV radiation, and the rate in which tanning gradually decreases after reaching its peak, depend on a person's skin tone, the type of UV radiation (A, B or both) and other biological characteristics. While it is not possible to identify the precise process at the individual level, the literature guides us to use the average UV radiation in the preceding three weeks [24].

After merging the datasets, our final dataset comprises 1,797,791 respondent-by-week observations of 4,020 individuals that were followed from 2000 to 2015. We use separate person fixed effects models for each skin tone category to predict the effects of the average UV radiation in the previous three weeks on the employment status of respondents. By doing so, we control for all the unobserved and time-invariant characteristics of the respondents in our data. Overall, we expected the effects on employment of the average UV radiation in the previous three weeks to be negative for the people with intermediate skin tones but not for those with the lightest and darkest skin tones.

## Results

Table 1 presents the descriptive statistics for the variables we use in the analysis.

We begin by documenting the correlation between one's skin tone and the probability of being employed. We estimate the following logistic regression model,

$$\ln \frac{p_{it}}{1 - p_{it}} = skintone_i \beta + \delta X_{it} + \varphi_t + \varepsilon_{it},$$

where $p_{it}$ is the probability that person $i$ at time $t$ is employed. The model controls for the year, week, region, education, occupation, industry, sex, marital status, age and parental status. We find that having a darker skin tone is associated with lower odds of employment. On average, having a skin tone that is one unit darker (on a scale of 1–10) leads to a 0.918 fold decrease in the odds of being employed (S1 Table in the S1 Appendix). Recall that the skin tone of respondents in the dataset was evaluated only once in 2008. Hence, for each respondent there is only one recorded skin tone. Therefore, the effects we report here are between and not within respondents. To better understand the magnitude of the results, we calculate the marginal effects corresponding to the odds ratios reported in the table: having a skin tone that is one unit darker, generated a 0.0047 decrease in the probability of being employed (p<0.001).

We now turn to explore the effects that exposure to UV radiation has on one's assessed skin tone in our dataset. We wish to know whether indeed greater UV radiation makes people look darker. The skin tones of respondents were assessed (on different dates in 2008) on a scale of 0–10. We want to test whether in our dataset, people whose skin tones were assessed in days with greater UV radiation, were indeed assessed as having darker skin tones.

Unfortunately, participants' skin tones were assessed during the fall, winter and spring of 2008, but not during the summer (when the effects of UV radiation on one's skin tone are expected to be the greatest). Thus, we cannot fully estimate the magnitude of the effects of UV radiation on the assessments of skin tone. Instead, we report the effects of UV radiation on the assessed skin tone for the dates available in our data set (fall, winter and spring of 2008). We

**Table 1. Descriptive statistics.**

| | Lightest Tones | | | Intermediate Tones | | | Darkest Tones | | | All | |
|---|---|---|---|---|---|---|---|---|---|---|---|
| | Mean | S.D. | N | Mean | S.D. | N | Mean | S.D. | N | Min | Max |
| Employed | 0.94 | 0.23 | 461,649 | 0.91 | 0.28 | 1,103,803 | 0.87 | 0.34 | 272,496 | | |
| UV (average, past 3 weeks) | 4.92 | 2.82 | 461,649 | 5.16 | 2.82 | 1,103,803 | 5.42 | 2.78 | 272,496 | 0.06 | 12.63 |
| Year | 2008.23 | 4.16 | 461,649 | 2008.22 | 4.21 | 1,103,803 | 2008.22 | 4.25 | 272,496 | 2000 | 2015 |
| Region: | | | | | | | | | | | |
| Midwest | 0.19 | 0.39 | 461,649 | 0.2 | 0.4 | 1,100,835 | 0.12 | 0.32 | 271,625 | | |
| Northeast | 0.26 | 0.44 | 461,649 | 0.21 | 0.4 | 1,100,835 | 0.16 | 0.36 | 271,625 | | |
| South | 0.28 | 0.45 | 461,649 | 0.33 | 0.47 | 1,100,835 | 0.65 | 0.48 | 271,625 | | |
| West | 0.27 | 0.44 | 461,649 | 0.26 | 0.44 | 1,100,835 | 0.07 | 0.26 | 271,625 | | |
| Female | 0.5 | 0.5 | 461,649 | 0.5 | 0.5 | 1,103,803 | 0.45 | 0.5 | 272,496 | | |
| White | 0.85 | 0.35 | 461,649 | 0.35 | 0.48 | 1,103,803 | 0.04 | 0.2 | 272,496 | | |
| Black | 0.01 | 0.08 | 461,649 | 0.31 | 0.46 | 1,103,803 | 0.91 | 0.29 | 272,496 | | |
| Hispanic | 0.14 | 0.34 | 461,649 | 0.33 | 0.47 | 1,103,803 | 0.04 | 0.2 | 272,496 | | |
| Age | 26.29 | 4.21 | 461,649 | 26.28 | 4.28 | 1,103,803 | 26.3 | 4.32 | 272,496 | 18 | 35.92 |
| Married | 0.3 | 0.46 | 460,236 | 0.25 | 0.43 | 1,101,712 | 0.17 | 0.38 | 272,164 | | |
| Children in HH | 0.42 | 0.81 | 461,186 | 0.67 | 1.04 | 1,102,171 | 0.77 | 1.15 | 271,986 | 0 | 12 |
| Education: | | | | | | | | | | | |
| Less than High School | 0.14 | 0.35 | 461,649 | 0.21 | 0.41 | 1,103,803 | 0.26 | 0.44 | 272,496 | | |
| High school | 0.43 | 0.5 | 461,649 | 0.5 | 0.5 | 1,103,803 | 0.57 | 0.49 | 272,496 | | |
| College | 0.43 | 0.49 | 461,649 | 0.29 | 0.45 | 1,103,803 | 0.17 | 0.38 | 272,496 | | |
| Skin Tones: | | | | | | | | | | | |
| 0 | 0.03 | 0.17 | 461,649 | 0 | 0 | 1,103,803 | 0 | 0 | 272,496 | | |
| 1 | 0.97 | 0.17 | 461,649 | 0 | 0 | 1,103,803 | 0 | 0 | 272,496 | | |
| 2 | 0 | 0 | 461,649 | 0.4 | 0.49 | 1,103,803 | 0 | 0 | 272,496 | | |
| 3 | 0 | 0 | 461,649 | 0.25 | 0.43 | 1,103,803 | 0 | 0 | 272,496 | | |
| 4 | 0 | 0 | 461,649 | 0.13 | 0.33 | 1,103,803 | 0 | 0 | 272,496 | | |
| 5 | 0 | 0 | 461,649 | 0.12 | 0.32 | 1,103,803 | 0 | 0 | 272,496 | | |
| 6 | 0 | 0 | 461,649 | 0.11 | 0.32 | 1,103,803 | 0 | 0 | 272,496 | | |
| 7 | 0 | 0 | 461,649 | 0 | 0 | 1,103,803 | 0.43 | 0.5 | 272,496 | | |
| 8 | 0 | 0 | 461,649 | 0 | 0 | 1,103,803 | 0.38 | 0.48 | 272,496 | | |
| 9 | 0 | 0 | 461,649 | 0 | 0 | 1,103,803 | 0.15 | 0.35 | 272,496 | | |
| 10 | 0 | 0 | 461,649 | 0 | 0 | 1,103,803 | 0.04 | 0.2 | 272,496 | | |
| Unique Individuals | | 1,064 | | | 2,373 | | | 583 | | | |
| Sun Occupation Index | 28.83 | 25.38 | 288,746 | 29.31 | 26.14 | 661,107 | 32.74 | 26.79 | 154,618 | 0 | 100 |
| Unique Individuals | | 623 | | | 1,329 | | | 300 | | | |
| Employment Duration (in Occ) | 7.00 | 2.60 | 408,665 | 6.75 | 2.53 | 957227 | 6.34 | 2.57 | 226,447 | 0.24 | 26 |
| Unique Individuals | | 878 | | | 1,826 | | | 405 | | | |

also report the effects of UV radiation in those non-summer days in which the UV radiation levels were relatively high. To get a sense of the magnitude of the effects of UV radiation on one's assessed skin tone during the summer (which is unavailable in our data set) we test for the effects of UV radiation on the one's assessed skin tone on those non summer days with relatively high UV radiation that are available in our data set. We treat days as high UV radiation days if the UV radiation in the previous three weeks was greater than 2 standard deviations below the average UV radiation (in the previous three weeks) in the summer. These estimates serve as a lower bound for the effects of UV radiation on one's perceived skin tone during the summer (which are unavailable in our data set).

We evaluate the effects of the average UV radiation in the previous three weeks on one's assessed skin tone in OLS regression models predicting the assessed skin tone of respondents. We estimate the following linear regression model,

$$Skintone_{it} = \alpha + UV_{it}\beta + \delta X_{it} + \varepsilon_{it}$$

The model controls for race and gender. In some specifications we further controls for week, state, industry and occupation. We report the results in S2 Table in the S1 Appendix. We find that indeed when respondents' skin tones were evaluated when UV radiation was greater, their skin tones were perceived to be darker. The effects reported in the models are lower bounds as participants' skin tones were not assessed in the summer when the effects of UV radiation on one's skin tone is expected to be the greatest (and to accumulate). Indeed, in our data the effects of UV radiation on one's assessed skin tone are greater when the assessment was done when UV levels are higher (model 3, $p<0.05$).

Finally, we expect the effects of the average UV radiation in the previous three weeks on the assessed skin tone to be weaker for people with very light or very dark skin tones. In models 4,5 and 6 (S2 Table in the S1 Appendix), we therefore report the results of OLS regression models predicting the effects of the average UV radiation in the previous three weeks on the assessed skin tone only on the subsample of respondents with skin tones of 2–8. To make sure that the respondents with intermediate skin tones who were perceived by the evaluators to be darker (because of exposure to greater UV radiation in the previous three weeks) are also included in the sample used for these models, we use a subsample of respondents with skin tones of 2–8 and not only those with skin tones of 2–6.

Indeed, we find that effects on this subsample are greater than the effects on the entire population. In Model 6—our preferred specification for a lower bound–we find that a one-unit increase in the UV radiation in the previous three weeks increases the assessed skin tone by 0.49 (Model 6).

Note that it is unclear what determined the interview dates for respondents and it may be that assignment was not random. In order to deal with a possible selection, in models 2, 3, 5 and 6 we control for the state, the week, the industry, the occupation and the race of respondents. In sum, these results suggest that exposure to UV radiation makes people seem as if they have a darker skin tone.

We now turn to our main analysis of the effects of UV radiation on respondents' probability of employment. We estimate person fixed effects logistic regression models predicting the employment status of respondents, for each of the three skin-tone categories,

$$\ln \frac{p_{it}}{1 - p_{it}} = \alpha + UV_{it}\beta + \delta X_{it} + \gamma_i + \varphi_t + \varepsilon_{it},$$

In the models we control for the year, the week, the region, as well as one's occupation (at the 2-digit Standard Occupational Classification), industry (at the 2-digit Standard Industrial Classification) and demographic characteristics. In Table 2, we present the results (odds ratios). Standard errors are clustered by metropolitan areas (stations). Recall that the models we use are all person fixed-effects models so that effects are estimated within respondents (i.e., the effect of the average UV radiation in the previous three weeks is estimated within person). Standard errors are clustered by geographical regions. Biases are corrected (for id and week) using an analytical bias correction derived by Fernandez-Val and Weidner (2016) [25]. In all the models we report in the paper (Tables 2–3 and S3-S5 Tables in the S1 Appendix) standard errors are clustered by region and biases are analytically corrected [25]. Clustering the standard errors by metropolitan areas generated similar significance levels. Note that in Tables 2

**Table 2. Logistic regression models predicting employment (Odds Ratios).**

| | Lightest Tones | | Intermediate Tones | | Darkest Tones | |
|---|---|---|---|---|---|---|
| | **(1)** | **(2)** | **(3)** | **(4)** | **(5)** | **(6)** |
| UV | 0.980 | 1.005 | 0.974*** | 0.960*** | 1.010 | 1.010 |
| Person Fixed Effects | Y | Y | Y | Y | Y | Y |
| Week, Year and Region Dummies | Y | Y | Y | Y | Y | Y |
| Demographics: Age, Married, Children in HH | | Y | | Y | | Y |
| Education Dummies | | Y | | Y | | Y |
| Occupation Dummies | | Y | | Y | | Y |
| Industry Dummies | | | | Y | | Y |
| Pseudo R2 | 0.228 | 0.202 | 0.256 | 0.21441 | 0.235 | 0.2044 |
| N | 340,061 | 323,634 | 888,275 | 807,827 | 236,708 | 213,619 |
| UV (log odds ratio) | -0.020 | 0.005 | -0.027*** | -0.041*** | 0.010 | 0.010 |
| | (0.011) | (0.012) | (0.006) | (0.007) | (0.011) | (0.011) |

*** = 0.001

** = 0.01

* = 0.05; Standard errors, in parentheses, are clustered by metropolitan areas (regions).

and 3 we report both the odds ratios and the log odds ratios. In the appendix we report only the odds ratios.

As expected, we find that for respondents with intermediate skin tones (those who tend to tan)—an increase in the average UV radiation in the previous three weeks results in a reduction in the likelihood of being employed, under both specifications. On average, a one unit change in the average UV radiation in the previous three weeks leads to a 0.96 fold decrease in the odds of being employed (p<0.001). As predicted our analysis suggests that for respondents

**Table 3. Logistic regression models predicting employment, by gender (for People with Intermediate Skin Tones, Odds Ratios).**

| | Men | Women |
|---|---|---|
| | **(1)** | **(2)** |
| UV | 0.913*** | 1.019 |
| Person Fixed Effects | Y | Y |
| Week, Year and Region Dummies | Y | Y |
| Demographics: Age, Married, Children in HH | Y | Y |
| Education Dummies | | Y |
| Occupation Dummies | Y | Y |
| Industry Dummies | Y | Y |
| Pseudo R2 | 0.212 | 0.228 |
| N | 405,032 | 407,300 |
| UV (log odds ratio) | -0.091 | 0.0185 |
| | (0.010) | (0.010) |

Standard errors are in parentheses

*** = 0.001

** = 0.01

* = 0.05

education dummies are not included for men because of linear dependence

who do not tend to tan (those with the lightest and darkest skin tones), changes in the average UV radiation in the previous three weeks do not affect the probability of employment.

To better understand the magnitude of the results, we calculate the marginal effect of the UV radiation in one's area in the previous three weeks controlling for other independent variables, on the probability of her employment: In Model 4, an increase of one unit in the average UV radiation in the previous three weeks, generated a 0.0021 decrease in the probability of being employed for participants with intermediate skin tones (and who therefore tend to tan) ($p < 0.001$). To put these effects in context, note that the average standard deviation in the UV radiation within week and location is between 0.21 units (in the winter) and 0.87 (in the summer). For example, in weeks 28–30 (summer time) in the JFK station (in NY) there was a 1.89 unit change in the average UV radiation between 2005 and 2006 (5.84 in 2005 compared to 7.73 in 2006). Thus, the changes in the UV radiation between the summer of 2005 and the summer of 2006 in the JFK station generated a (1.89*0.0021 =) 0.004 decrease in the probability of being employed for people with intermediate skin tones.

We now turn to compare the effects we observe in Table 2 to the effect that one's skin tone has on one's probability of being employed. Recall that in our data set (S1 Table in the S1 Appendix), having a darker skin tone is associated with lower odds of employment. When we calculate the marginal effect of one's skin tone on employment, we find that on average, having a skin tone that is darker by one unit (on a scale of 1–10) decreases the probability of being employed by 0.0047. Thus, the effects we observe in Table 2 of a one-unit change in the UV radiation in the previous three weeks on the probability of being employed equals about (0.0021/0.0047) 45% of the effect of having a skin tone that is one unit darker.

In order to make sure that the effects we find of UV radiation on employment are of reasonable magnitude (model 4, Table 2, a marginal effect of 0.0021), we evaluate them in light of our estimates of the effects of the UV radiation on the respondents' assessed skin tones (model 6, S2 Table in the S1 Appendix, an effect of 0.49) and of the effects of respondents' skin tones on their probability of being employed (S1 Table in the S1 Appendix, a marginal effect of 0.0047). In the light of these findings, our main findings seem reasonable in size (0.49*0.0047>0.0021).

Note that this is only done in order to make sure that the magnitude of the effects we observe are reasonable; Unfortunately, participants' skin tones were assessed during the fall, winter and spring of 2008, but not during the summer (when the effects of UV radiation on ones' skin tone are expected to be the greatest). 0.0047 is therefore a lower bound.

Because the sample size of respondents with intermediate skin tone in our data is significantly larger than the sample sizes of respondents with lightest and darkest skin tones, we rerun the analysis on a randomly selected small subsample of respondents with intermediate skin tones. We do so to make sure that it is not the differences in the sample sizes that generate statistically significant negative effects of UV radiation on employment only for people with intermediate skin tones. Out of the 2373 respondents with intermediate skin tones, we randomly selected a group of 583 (301,231 observations). Indeed, we find that even with the smaller randomly-selected subsample, an increase in the average UV radiation in the previous three weeks results in a reduction in the likelihood of being employed. We replicated the analyses on different randomly-selected subsamples of respondents with intermediate skin tones. The effects obtained were similar in size and statistical significance.

## Women and men

In the models presented in Table 3, we explore the differences between the effects of the average UV radiation in the previous three weeks on women and on men (with intermediate skin tones).

We find that the effect of the average UV radiation in the previous three weeks is negative for men but not for women. On average, a one unit change in the average UV radiation in the previous three weeks leads to a 0.913 fold decrease in the odds of being employed for male respondents with intermediate skin tones (and who therefore tend to tan) (p<0.001). One possible explanation for this finding is that it is discrimination against men that drives the results we observe. This possibility is consistent with the results of previous studies that have shown that race inequality in the American labor force is driven by differences between white and black men [19, 26, 27]. Another possible explanation is that women may be more likely than men to use sunscreen. Indeed, some studies suggest that women tend to use more sunscreen on the face than men. However, these studies also show that most adults (women and men) do not regularly use sunscreen on the face [28]. In any case, note that if a large number of respondents (women and men) in our dataset tended to use sunscreen, we would be underestimating the effects of UV radiation on employment in our analyses. As expected, the effects of the average UV radiation in the previous three weeks on the employment of men with the lightest and darkest skin tones are statistically non-significant.

## Sun exposure

One possible alternative explanation to our findings is that the people with intermediate skin tones tend to work in occupations in which employment is lower when UV radiation is higher. To rule out this explanation, in S3 Table in the S1 Appendix (Model 1), we include an Occupational Information Network's (O*NET) index variable that captures the degree to which one's occupation typically involves exposure to the sun (ranging from 1 to 100, at the 4-digit Standard Occupational Classification). In a separate analysis, we find that people with intermediate skin tones tend to work more in occupations that involve greater sun exposure. Because we found that effects are only significant for men, we present the results only for men.

We find that even after including the sun-exposure-index variable, the effect of the average UV radiation in the previous three weeks remains negative and significant for people with intermediate skin tones and non-significant for all others. We also find that the interaction between the sun exposure index variable and the average UV radiation in the previous three weeks is actually positive. This suggests that men are more likely, not less, to be employed in occupations that involve greater exposure to the sun when the average UV radiation increases. We therefore conclude that it is not the tendency of people with intermediate skin tones to work in occupations that involve exposure to the sun that drives our results.

## Place of residence

Another alternative explanation for the results we find is that the average UV radiation in the previous three weeks is correlated with one's place of residency that in turn affects employment in ways that are not observable in our dataset (and that are relevant only to people with intermediate skin tones). One such possible explanation might be that people with intermediate skin tones move to sunny places in which they tend to be employed less. To rule out this explanation, we report results of a logistic regression model predicting the effects of the average UV radiation in the previous three weeks on one's employment for the sample of men who have not yet changed their place of residency. By doing so, we keep respondents' place of residence constant. We find that even when the sample includes only observations in which the place of residence is constant, the effect of the average UV radiation in the previous three weeks on one's probability of employment is negative and statistically significant although much smaller; On average, a one unit change in the average UV radiation in the previous three weeks leads to a 0.932 fold decrease in the odds of being employed (p<0.001; Model 2, S3

Table in the S1 Appendix). This is a smaller effect compared to the effect of UV radiation in the previous three weeks on employment for all the men in our sample (0.913, p<0.001, Model 1, Table 3).

## Occupations with greater turnover

The design of our study enables us to capture discrimination only when people's employment statuses change. In other words, the more people's employment statuses change, the easier it would be for us to detect the effects of the average UV radiation in the previous three weeks on the probability of being employed. Therefore, we expect the effect of the UV radiation in the previous three weeks to be more detectible when respondents are employed in occupations with greater turnover. In Model 3 (S3 Table in the S1 Appendix), we therefore include an index variable capturing the duration of employment per occupation. This variable was constructed from the Current Population Survey (CPS) as the average duration of employment in the current position (in years), by occupation (ranging from 0.24 to 26, by year, at the 4-digit Standard Occupational Classification). We find that indeed, the effect observed in our analysis for the average UV radiation in the previous three weeks is greater when the duration of employment within the occupation is shorter. Note also that occupations with greater turnover tend to be low-status occupations. Indeed, in our data effects were stronger for low-status occupations. This may be because of the above-mentioned effect of turnovers but also because with low status occupations the length of time between interview and employment is shorter compared to with high status occupations (where oftentimes the interview takes place weeks before the change in employment status happens). In other words, with high status occupations (compared to low-status occupations) looking darker when first employed is less strongly correlated with being darker when interviewed. Thus, the design of our study makes it harder to detect discrimination on the basis of one's perceived skin tone in high-status occupations compared to low status occupations.

## Robustness checks and additional inquires

To check the robustness of the results we report, we try alternative specifications to test our hypothesis (S4 Table in the S1 Appendix). We use the average UV radiation in the previous four weeks (instead of the previous three weeks) (model 1); we use different categorizations of respondents' skin tones (models 2–4); and we test for nonlinear effects of the average UV radiation in the previous three weeks (model 5). In all alternative specifications, the results obtained are essentially the same with negative and statistically significant effects of the average UV radiation on the probability of employment for respondents with intermediate skin tones, but not others.

We also estimate the effects of longer lags of UV radiation. We find that other specifications are also negatively correlated with the current employment of people with intermediate skin tones. As predicted, for people with lighter or darker skin tones, effects on employment are statistically non-significant (models are estimated on male non movers whose skin tones were assessed in days in which the average UV radiation in the previous three weeks was relatively high).

In addition, we explore the effects of the season in which respondents' skin tone was assessed on the results we obtain. Recall that the skin tones of respondents were assessed on different dates in 2008. Whereas the skin tone of many respondents was assessed in the winter, the skin tone of some was assessed in May or in October when UV radiation tends to be greater (none were interviewed during the June-September period). Because some skin tones are affected by UV radiation, we worry that the assessed skin tones of the respondents that

were interviewed in May or October are darker than they would have been if assessed in the winter, generating inaccurate classification of the three skin tone categories we use. More specifically, we are concerned that those assessed in May and October as having intermediate skin tones, actually have lighter skin tones than those assessed as having intermediate skin tones in the winter.

In Models 1 and 2 presented in S5 Table in the S1 Appendix, we therefore separate those respondents (with intermediate skin tones) whose skin tone was assessed in the winter from those assessed in May or October. Indeed, we find that the general effects for the average UV radiation in the previous three weeks we observe are driven by the people whose skin tone was assessed in the winter, and not in May or October where exposure to UV radiation might have affected the assessment of skin tone.

We further explore the effects on employment of the average UV radiation in the future (weeks 5–7) together with the average UV radiation in the previous three weeks (on people with intermediate skin tones), as a falsification test (Model 3, S5 Table in the *S1 Appendix*). We find that it is only the UV radiation in the previous three weeks but not in the following three weeks that affects one's employment. We also estimate the effects of ten data points of UV radiation (weeks 1–10) in the future on past employment outcomes (together with the average UV radiation in the previous three weeks; on people with intermediate skin tones). All ten future data points estimated had statistically non-significant effects on past employment. Next, we estimate the effects of the average UV radiation in the previous three weeks within race. We find that even within race, greater UV radiation in the previous three weeks is associated with lower probability of employment (Models 4–6, S5 Table in the *S1 Appendix*).

In model 4, we see that even within whites, the effects of UV radiation on employment are negative and statistically significant. Recall that in our sample (Table 1) 35% of the people with intermediate skin tones were whites. These findings suggest that even within this white population looking darker is associated with a lower probability of being employed. This highlights the negative effects of colorism even within race, and perhaps surprisingly even within whites.

Estimations of time trends and variations by geographical regions are all statistically non-significant.

Note that concerns of reversed causality (i.e, that unemployed individuals spend more time in the sun and thus look darker) are irrelevant for our research design; we do not observe how dark people currently look nor how much time they actually spend in the sun. We only proxy their potential to look darker by using their assessed skin tone in 2008 and the (exogenously given) average UV radiation in the previous three weeks. In other words, being unemployed in the present cannot affect the assessed skin tone in 2008 or the average UV radiation in the previous three weeks.

Moreover, in our data set, the assessed skin tones of unemployed participants are not affected more by the average UV radiation in the previous three weeks than the assessed skin tones of the employed participants. In models predicting the assessed skin tone by the average UV radiation in the previous three weeks (similar to the models presented in S2 Table in the S1 Appendix), an interaction between being unemployed and the average UV radiation in the previous three weeks is statistically non-significant.

Finally, an additional set of possible concerns would be that people who look like they spent time in the sun are perceived to be lazier or less committed to the labor force than people who do not, and that the effects we observe are the results of these perceptions. Note however, that people with intermediate skin tones (unlike those with very light skin tones) look as if their skin tone is originally darker after spending time in the sun (see the results presented in S2 Table in the S1 Appendix). More importantly, there is no reason to assume that only people

with intermediate skin tones who spend time in the sun are perceived to be lazy or less committed, but not people with the lightest skin tones who spend time in the sun.

Interestingly, one recent study has shown that when the weather is nicer, workers are more likely to report being sick [29]. This should not bias our results however, because in the NLSY97 dataset sickness or absenteeism are not coded as unemployment.

## Conclusion

In this paper we present the results of a natural experiment—we use people's tendency to tan when exposed to UV radiation to test for employment discrimination on the basis of skin tone. We show that UV radiation negatively affects the likelihood of being employed for people whose skin tone becomes darker by exposure to the sun, but not for others. These within-person findings hold even when controlling for the week, the year, the region, demographic characteristics and the industry and occupation one is employed in and when place of residency is held constant. Whereas various previous studies have documented racial inequalities in the American labor force, as well as inequalities on the basis of skin tone (even within race), discrimination has been very hard to prove.

The main contribution we make here is by providing evidence for a causal effect between one's skin tone and one's probability of being employed. We thus document labor force discrimination—and not merely inequalities—on the basis of one's skin tone. By focusing on the tendency to tan, we do not imply that the experiences associated with race and racism can be reduced to the experiences associated with tanning. What we wish to do here is to focus on the discriminatory practices of employers–who oftentimes cannot distinguish between those who look darker because they spent time in the sun and those with naturally darker skin tones.

## Supporting information

**S1 Appendix.**
(DOCX)

## Acknowledgments

For insightful comments we are grateful to Ian Ayers, Adam Chilton, Alma Cohen, Eddie Dekel, Yehontan Givati, William Hubbard, Ariel Porat and the participants of the Chicago Law School Faculty Seminar, Cornell Law School Faculty Seminar, Northwestern Law School faculty seminar and the Tel-Aviv University Law and Economics Workshop. Dror Avidor provided superb research assistance. This research was conducted with restricted access to Bureau of Labor Statistics (BLS) data. The views expressed here do not necessarily reflect the views of the BLS.

## Author Contributions

**Conceptualization:** Tamar Kricheli Katz, Tali Regev.

**Data curation:** Haggai Porat.

**Formal analysis:** Tamar Kricheli Katz, Tali Regev, Haggai Porat.

**Investigation:** Tamar Kricheli Katz, Tali Regev, Haggai Porat, Ronen Avraham.

**Methodology:** Tamar Kricheli Katz, Tali Regev, Shay Lavie, Ronen Avraham.

**Project administration:** Tamar Kricheli Katz, Tali Regev.

**Resources:** Ronen Avraham.

**Supervision:** Tamar Kricheli Katz.

**Writing – original draft:** Tamar Kricheli Katz, Tali Regev, Shay Lavie.

**Writing – review & editing:** Tamar Kricheli Katz, Tali Regev, Shay Lavie, Haggai Porat, Ronen Avraham.

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
