## [Decision Letter · Decision Letter 0]

26 Mar 2020

PONE-D-20-03529

Those Who Tan and Those Who Don’t: A Natural Experiment on Colorism

PLOS ONE

Dear Dr. Kricheli-Katz,

Thank you for submitting your manuscript to PLOS ONE. After careful consideration, we feel that it has merit but does not fully meet PLOS ONE’s publication criteria as it currently stands. Therefore, we invite you to submit a revised version of the manuscript that addresses the points raised during the review process.

Both the qualified reviewers are labor economists and they like the novelty of your research. I myself also enjoyed reading your paper. But the reviewers also raised some major concerns. Please try to address their concerns as much as you can. In addition, I also have some comments and suggestions.

First, reviewer 1 is a bit confused about the coefficients versus odds ratio in your table. I suggest you at least report both the coefficients (log odds ratio) and odds ratio in Table 2 and then you can state you will report only odds ratio in the rest of tables. By looking at the negative coefficients for column 3 and 4 of table 2, readers will see your results intuitively.

Second, as reviewer 2 pointed out women may be more likely to use sunscreen. Such kind of avoidance behavior may occur among men too. For example, if people are concerned about the risk of skin cancer, they may be more likely to avoid sunshine on days with high degree of UV radiation. If this is true also for people with intermediate tones, the effect of UV on employment is biased toward zero. This may be true if people with intermediate tone understand they may be more likely to be discriminated if their skin get darker.

Reviewer 2 also raised a valid question on the tradeoff between nice weather and leisure or work. Actually there is a study showing that when weather is nice workers are more likely to report sick absence (Shi and Skuterud, 2015, Gone Fishing! Reported Sickness Absenteeism and the Weather, Economic Inquiry 53(1), 388-405.) You may cite this paper. One way to address this concern is to check full-time workers versus part time workers. UV radiation is expected to affect full time employment less but part time employment more if people do value leisure on sun shining days. In this case you may use an ordered logit model, using 0 as unemployment, 1 as part time, 2 as full time.

Third, the within-race results in columns 4-6 of Table A4 is tricky to interpret. On the one hand, they show your estimation results are robust; on the other hand, why would the UV effect hold for white people assuming white people are not discriminated and have light tone. Does Column 4 suggest a “tan” effect—people who got a tan are more likely to be unemployed? Since your models have included individual fixed effects which absorb the racial effect, I tend to suggest you drop these columns unless you have a better explanation for the white sample.

One minor question out of my curiosity is how many days does it take to get a tan and how long does a tan persist? Are there any medical or scientific studies on these questions?

We would appreciate receiving your revised manuscript by May 02 2020 11:59PM. To enhance the reproducibility of your results, we recommend that if applicable you deposit your laboratory protocols in protocols.io, where a protocol can be assigned its own identifier (DOI) such that it can be cited independently in the future. For instructions see: http://journals.plos.org/plosone/s/submission-guidelines#loc-laboratory-protocols

We look forward to receiving your revised manuscript.

Kind regards,

Shihe Fu, Ph.D.

Academic Editor

PLOS ONE

Journal Requirements:

3.  We note that Figure [1 A] in your submission contain copyrighted images. All PLOS content is published under the Creative Commons Attribution License (CC BY 4.0), which means that the manuscript, images, and Supporting Information files will be freely available online, and any third party is permitted to access, download, copy, distribute, and use these materials in any way, even commercially, with proper attribution. For more information, see our copyright guidelines: http://journals.plos.org/plosone/s/licenses-and-copyright.

1.         You may seek permission from the original copyright holder of Figure [1 A] to publish the content specifically under the CC BY 4.0 license. 

Reviewers' comments:

Reviewer's Responses to Questions

**Comments to the Author**

1. Is the manuscript technically sound, and do the data support the conclusions?

Reviewer #1: Partly

Reviewer #2: Yes

2. Has the statistical analysis been performed appropriately and rigorously? 

Reviewer #1: Yes

Reviewer #2: Yes

3. Have the authors made all data underlying the findings in their manuscript fully available?

Reviewer #1: Yes

Reviewer #2: Yes

4. Is the manuscript presented in an intelligible fashion and written in standard English?

Reviewer #1: Yes

Reviewer #2: Yes

5. Review Comments to the Author

Reviewer #1: Referee Report

PONE-D-20-03529

Those Who Tan and Those Who Don’t: A Natural Experiment on Colorism

Summary:

This paper looks for the evidence of colorism by studying the effect of skin tone on individual’s employment. Combining individual-level weekly employment data from NLSY97 with the data of UV radiation collected from weather stations, the paper explores whether the exposure to UV radiation during the preceding three weeks affects an individual’s employment. In particular, the main results are obtained for three skin tone categories separately (light, intermediate, dark). Based on the logistic regression model controls for individual fixed effects, the effect is only found statistically significant for the intermediate skin tone category, who are most likely to get darker under UV radiation.

General Comments:

This paper novelly exploits the weekly variations in UV radiation, which potentially shifts an individual’s skin tone, and individual’s employment to empirically test colorism. Moreover, instead of basing on cross-sectional variations, this paper increases credibility by using within-individual variations and examines the heterogeneous effects by different skin tone categories. This paper is also rich in details, taking care of several relevant concerns. Nevertheless, there are several crucial mistakes or confusions needed to be clarified before drawing convincing conclusions.

a. The most concerning and confusing results are the main ones reported in Table 2. The paper claims that UV radiation reduces the likelihood of being employed. However, all coefficients of UV are positive. If the effect is indeed negative, there are three possibilities to explain the results here: (1) dependent variable is an indicator of unemployment instead of employment. (2) Greater value in the UV variable associates with lower UV radiation level. (3) The estimates are negative but the authors report them with typos. The first possibility is against the results in Table 4. For Table 4, the authors claim in page 18 that people “tend to work more in occupations that involve greater sun exposure”, which suggests indeed the dependent variable equals 1 for employment instead of unemployment. The second possibility is against the description of the UV variable in page 6. The third possibility is unlikely as these “typos” exist in every columns in Table 2, Table 3, Table4 as well as Table A1. If these were indeed typos, mistakes are grave and they undermine the rigor of this work.

b. Please report standard errors for all coefficients instead of only indicating the significance level, especially for Table 2 and Table3. This is quite important as the main conclusion that UV has negative effect only for the intermediate tones relies heavily on the statistical significance. Examining the results in Table 2, actually coefficients have close magnitudes for all three skin tone categories.

c. Still related to the similar magnitudes of coefficients in all three skin tone categories, the difference in sample size across categories is a concern: intermediate tones have significantly larger sample size. While the authors claim that the same results are found for a random smaller sample for the intermediate tones, please report the numerical results (the coefficients, the sample size, and the standard errors).

d. Coefficients of almost all variables, regardless of the significance, have the magnitude between 0.9 and 1.0 in the regressions of employment. Consider that different variables have quite different units and scales, such as the variables UV, UV^2 and UV^3 in Table A3, or the variable Sun Exposure Index in Table 4, it is unsure whether this is just a coincidence or not (Are these variables normalized? ). The paper suggests that the main results are based on the individual fixed-effects logit model. Is this the conditional logit model, or just a standard logit regression with individual dummies? The latter one may suffer from the issue of incidental parameters and the estimates are inconsistent. For easier and clearer interpretations, can the authors report results based on the linear probability model with individual fixed effects (at least for Table 2)?

Other Minor Comments:

a. How many individuals are included in the sample? Page 6 suggests the number is 4020, while the descriptive statistics in Table 1 by summing up the row Unique Individuals suggests less.

b. Related to the interpretation of the fixed-effects logit regressions, why the effect of Sun Exposure Index of each occupation can be identified in Table 4? Is this Index provided by O*NET longitudinal? To the best of my knowledge, O*NET provides cross-sectional information for each occupation. How is the effect of this time-invariant variable identified in the fixed-effects model?

c. How does this paper determine the occupation, and occupation-related variables such as the Sun Exposure Index and the Employment Duration, for those individuals unemployed? These variables should be unavailable if an individual is unemployed. Particularly, if the individual has changed occupations across weeks, is there weekly occupation information?

d. The row Week, State, Industry and Occupation Dummies in Table A2 may miss the Y for models (2) and (3).

e. (Pseudo) R-squares should be reported in all regressions.

f. Temperature may be a factor correlates with both the UV and individual’s employment.

Reviewer #2: Referee Report for Manuscript No. PONE-D-20-03529 “Those Who Tan and Those Who Don’t: A Natural Experiment on Colorism”

This manuscript provides evidence on the effect of one’s skin tone on his or her probability of being employed, use exposure to UV radiation as a natural experiment. The authors find male and those with intermediate skin tones are less likely to be employed when the UV radiation in the previous three weeks in the area in which they reside is greater. This is an interesting paper, and below are my comments.

Main comments:

1. The main question, as mentioned in the abstract, is “are darker-skinned workers discriminated against in the labor market?”. However, I don’t think the results found in the paper can fully separate discrimination from alternative hypotheses. For example, if greater UV radiation makes people want to enjoy life more and reduce their job search effort or work effort in the labor market, we would find the same negative correlation between UV radiation and employment probability. Although the authors did mention this possibility in the manuscript (last paragraph in page 23), they argue that only people with intermediate skin tones finds this significant negative effect, but not other people, can rule out the work effort hypothesis. I do not completely agree. It is still possible that intermediate skin tones are related to specific personal characteristics, and those are the people who will reduce their work effort when UV radiation is greater. I would not emphasize too much on a causal discrimination story, and would just tell an interesting correlation between UV radiation and employment, and acknowledge it could due to discrimination or some other reasons.

2. The authors find that UV radiation only has effect on men’s employment but not women’s, and interpret it as “this suggests that it is discrimination against men that drives the results we observe” (page 16, last sentence). Again, one can think of alternative stories other than discrimination. Perhaps women are more likely to use sunscreen and less likely to get darker. Or perhaps women drop out of labor force (e.g., stay home and take care of children) rather than become unemployed. In the latter case, maybe the authors can provide analysis on labor force participation, in addition to employment.

3. Does NLSY surveys every year and the weekly employment status is based on recall? Recalling employment status every week for the past year may not be accurate and is subject to large measurement error. The authors should discuss that.

4. During Christmas season or winter break, many people travel to other city or state and the UV radiation of where they usually reside (I assume the place of reside only reports once each survey year in NLSY) would have no effect on their skin tone. Since the data provide interview dates, perhaps the authors can exclude holiday season where people travel the most, as a robustness check. People can be out of town in other weeks too, but it maybe difficult to detect that.

Minor comments:

5. The authors conduct several solid regression analysis. I would suggest the authors to write out the regression equation explicitly before showing the results.

6. In addition to group the data into three skin tone categories (light, intermediate, dark), I would like to see results in finer categories and not to group them together (perhaps in the appendix), given the data have a large sample size.

7. Table A2 is not mentioned in the text, perhaps it should be referred to in the last paragraph of page 10? What’s the difference between column (1) and (2) in Table A2? Perhaps there is a missing “Y” in the column (2) for dummies, so does column (3)?

6. PLOS authors have the option to publish the peer review history of their article (what does this mean?). If published, this will include your full peer review and any attached files.

Reviewer #1: No

Reviewer #2: No

---

## [Author Response · Author response to Decision Letter 0]

5 May 2020

Dear Shihe Fu, 

Thank you very much for the valuable comments and suggestions. Following them, we revised the manuscript. Please see the updated manuscript (changes in ‘track changes’) and our responses to the comments below (in Italics). 

Please note that that there are legal restrictions on sharing the NLSY97 de-identified data set that contains detailed information on the geographic residence of each NLSY97 respondent. The Geocode CD is released only to those who satisfactorily complete the Bureau of Labor Statistics geocode agreement procedure.

The geocode application document is available online at www.bls.gov/nls/geocodeapp.htm

Because we link the UV radiation data to the geographic residence of respondents, we cannot share the data (but we can share the codes if needed). 

Sincerely, 

Tamar Kricheli Kat, Tali Regev, Shai Lavie, Haggai Porat and Ronen Avraham. 

Dear Dr. Kricheli-Katz,

Thank you for submitting your manuscript to PLOS ONE. After careful consideration, we feel that it has merit but does not fully meet PLOS ONE’s publication criteria as it currently stands. Therefore, we invite you to submit a revised version of the manuscript that addresses the points raised during the review process.

Both the qualified reviewers are labor economists and they like the novelty of your research. I myself also enjoyed reading your paper. But the reviewers also raised some major concerns. Please try to address their concerns as much as you can. In addition, I also have some comments and suggestions.

First, reviewer 1 is a bit confused about the coefficients versus odds ratio in your table. I suggest you at least report both the coefficients (log odds ratio) and odds ratio in Table 2 and then you can state you will report only odds ratio in the rest of tables. By looking at the negative coefficients for column 3 and 4 of table 2, readers will see your results intuitively.

Following this recommendation, we now report both the odds ratio and the log odds ratio in Tables 2 and 3 and then we state that we report only odds ratio in the appendix tables. 

Second, as reviewer 2 pointed out women may be more likely to use sunscreen. Such kind of avoidance behavior may occur among men too. For example, if people are concerned about the risk of skin cancer, they may be more likely to avoid sunshine on days with high degree of UV radiation. If this is true also for people with intermediate tones, the effect of UV on employment is biased toward zero. This may be true if people with intermediate tone understand they may be more likely to be discriminated if their skin get darker.

Thank you for your comment. We now include this alternative explanation in the text. We note however that studies have shown that most adults do not use sunscreen on the face (page 18). 

Reviewer 2 also raised a valid question on the tradeoff between nice weather and leisure or work. Actually there is a study showing that when weather is nice workers are more likely to report sick absence (Shi and Skuterud, 2015, Gone Fishing! Reported Sickness Absenteeism and the Weather, Economic Inquiry 53(1), 388-405.) You may cite this paper. One way to address this concern is to check full-time workers versus part time workers. UV radiation is expected to affect full time employment less but part time employment more if people do value leisure on sun shining days. In this case you may use an ordered logit model, using 0 as unemployment, 1 as part time, 2 as full time.

Thank you for the reference. We include it in the paper and discuss it in the text (page 24). Note however that we were not clear enough in explaining that in our dataset sickness or absenteeism are not coded as unemployment. This would mean that people who are absent are treated as employed in our analyses. For this reason, part time (that indeed involves more leisure) is not expected to affect the tendency to be unemployed in nice weather. We hope that the text is clearer now. 

Third, the within-race results in columns 4-6 of Table A4 is tricky to interpret. On the one hand, they show your estimation results are robust; on the other hand, why would the UV effect hold for white people assuming white people are not discriminated and have light tone. Does Column 4 suggest a “tan” effect—people who got a tan are more likely to be unemployed? Since your models have included individual fixed effects which absorb the racial effect, I tend to suggest you drop these columns unless you have a better explanation for the white sample.

Following this comment, we added an explanation to the model (4) in the text (page 23). Please Let us know if you still think we should remove the model (and we will). (Note that this is now Table A5, since we moved Table 3 to be A3 in the Appendix)

One minor question out of my curiosity is how many days does it take to get a tan and how long does a tan persist? Are there any medical or scientific studies on these questions?

There are no clear answers to this question. The design of our study builds on these two papers:

 Fitzpatrick, T. B. (1975). "Soleil et peau" [Sun and skin]. Journal de Médecine Esthétique (in French), (2), 33–34; 

Fitzpatrick, T.B. (1988). The validity and practicality of sun-reactive skin types i through vi. Archives of Dermatology, 124(6), 869–871. 

Because affects are unclear we also try the UV radiation in the previous 4 weeks for robustness (p. 21 and Table A4). 

Responses to the reviewers

Reviewer #1: Referee Report

PONE-D-20-03529

Those Who Tan and Those Who Don’t: A Natural Experiment on Colorism

Summary:

This paper looks for the evidence of colorism by studying the effect of skin tone on individual’s employment. Combining individual-level weekly employment data from NLSY97 with the data of UV radiation collected from weather stations, the paper explores whether the exposure to UV radiation during the preceding three weeks affects an individual’s employment. In particular, the main results are obtained for three skin tone categories separately (light, intermediate, dark). Based on the logistic regression model controls for individual fixed effects, the effect is only found statistically significant for the intermediate skin tone category, who are most likely to get darker under UV radiation.

General Comments:

This paper novelly exploits the weekly variations in UV radiation, which potentially shifts an individual’s skin tone, and individual’s employment to empirically test colorism. Moreover, instead of basing on cross-sectional variations, this paper increases credibility by using within-individual variations and examines the heterogeneous effects by different skin tone categories. This paper is also rich in details, taking care of several relevant concerns. Nevertheless, there are several crucial mistakes or confusions needed to be clarified before drawing convincing conclusions.

a. The most concerning and confusing results are the main ones reported in Table 2. The paper claims that UV radiation reduces the likelihood of being employed. However, all coefficients of UV are positive. If the effect is indeed negative, there are three possibilities to explain the results here: (1) dependent variable is an indicator of unemployment instead of employment. (2) Greater value in the UV variable associates with lower UV radiation level. (3) The estimates are negative but the authors report them with typos. The first possibility is against the results in Table 4. For Table 4, the authors claim in page 18 that people “tend to work more in occupations that involve greater sun exposure”, which suggests indeed the dependent variable equals 1 for employment instead of unemployment. The second possibility is against the description of the UV variable in page 6. The third possibility is unlikely as these “typos” exist in every columns in Table 2, Table 3, Table4 as well as Table A1. If these were indeed typos, mistakes are grave and they undermine the rigor of this work.

Thank you for this comment. Following your comment, we clarified the tables We now explain when odds ratios and when log odds ratios are used (the coefficients are positive in table 2 only when odds ratios are reported. Because they are smaller than one these are all negative effects). We hope that now the results are reported in a clearer way. 

b. Please report standard errors for all coefficients instead of only indicating the significance level, especially for Table 2 and Table3. This is quite important as the main conclusion that UV has negative effect only for the intermediate tones relies heavily on the statistical significance. Examining the results in Table 2, actually coefficients have close magnitudes for all three skin tone categories.

Following your comment, we added the standard errors to tables 2 and 3. 

c. Still related to the similar magnitudes of coefficients in all three skin tone categories, the difference in sample size across categories is a concern: intermediate tones have significantly larger sample size. While the authors claim that the same results are found for a random smaller sample for the intermediate tones, please report the numerical results (the coefficients, the sample size, and the standard errors).

We were not sure how to report the results for the small random samples (because they vary from one sample to another and the differences between them are not informative. We do report the sample sizes of the small sizes in the paper: “we randomly selected a group of 583 (301231 observations)”. 

d. Coefficients of almost all variables, regardless of the significance, have the magnitude between 0.9 and 1.0 in the regressions of employment. Consider that different variables have quite different units and scales, such as the variables UV, UV^2 and UV^3 in Table A3, or the variable Sun Exposure Index in Table 4, it is unsure whether this is just a coincidence or not (Are these variables normalized? ). The paper suggests that the main results are based on the individual fixed-effects logit model. Is this the conditional logit model, or just a standard logit regression with individual dummies? The latter one may suffer from the issue of incidental parameters and the estimates are inconsistent. For easier and clearer interpretations, can the authors report results based on the linear probability model with individual fixed effects (at least for Table 2)?

See our response to (a). Because we are reporting the odds ratios, the effects are all somewhere between 0.9 and 1 (but see the magnitudes and signs for the log odds ratios coefficients or the marginal effects). 

Other Minor Comments:

a. How many individuals are included in the sample? Page 6 suggests the number is 4020, while the descriptive statistics in Table 1 by summing up the row Unique Individuals suggests less.

Thank you for this comment. We do have 4020 unique respondents but there are some missing values for two variables in our data (turnover, exposure to the sun in the occupation). Following the comment, we updated table 1. We hope it better reflects the data now. 

b. Related to the interpretation of the fixed-effects logit regressions, why the effect of Sun Exposure Index of each occupation can be identified in Table 4? Is this Index provided by O*NET longitudinal? To the best of my knowledge, O*NET provides cross-sectional information for each occupation. How is the effect of this time-invariant variable identified in the fixed-effects model?

Indeed, the index provided by O*NET is not longitudinal. The effects are generated in the fixed effects models because some participants switched their occupations. 

c. How does this paper determine the occupation, and occupation-related variables such as the Sun Exposure Index and the Employment Duration, for those individuals unemployed? These variables should be unavailable if an individual is unemployed. Particularly, if the individual has changed occupations across weeks, is there weekly occupation information?

“The NLSY97 asks respondents age 14 or older to report their occupation for each employer. The question "what kind of work did you do" elicits information on the occupation when the job started. The occupational classification at the job's end date (or at the survey date for on-going jobs) is solicited for all employee jobs lasting more than 13 weeks. Survey staff then code the respondent's occupation at each job.”

This means that unemployed individuals are treated as having their previous occupations. 

d. The row Week, State, Industry and Occupation Dummies in Table A2 may miss the Y for models (2) and (3).

Indeed. Thank you. We updated Table A2. 

e. (Pseudo) R-squares should be reported in all regressions.

Following your comment, we added the (Pseudo) R-squares to all tables reporting logistic regression results, and (adjusted) R-squares to the table reporting the OLS results. 

f. Temperature may be a factor correlates with both the UV and individual’s employment.

Yes. This is probably true. Yet, there is no reason to assume that temperature affects only the employment of individuals with intermediate skin tones (only men actually). Also: the effects of temperature on employment are probably small (if any) when the week (and year) are controlled for. 

Reviewer #2: Referee Report for Manuscript No. PONE-D-20-03529 “Those Who Tan and Those Who Don’t: A Natural Experiment on Colorism”

This manuscript provides evidence on the effect of one’s skin tone on his or her probability of being employed, use exposure to UV radiation as a natural experiment. The authors find male and those with intermediate skin tones are less likely to be employed when the UV radiation in the previous three weeks in the area in which they reside is greater. This is an interesting paper, and below are my comments.

Main comments:

1. The main question, as mentioned in the abstract, is “are darker-skinned workers discriminated against in the labor market?”. However, I don’t think the results found in the paper can fully separate discrimination from alternative hypotheses. For example, if greater UV radiation makes people want to enjoy life more and reduce their job search effort or work effort in the labor market, we would find the same negative correlation between UV radiation and employment probability. Although the authors did mention this possibility in the manuscript (last paragraph in page 23), they argue that only people with intermediate skin tones finds this significant negative effect, but not other people, can rule out the work effort hypothesis. I do not completely agree. It is still possible that intermediate skin tones are related to specific personal characteristics, and those are the people who will reduce their work effort when UV radiation is greater. I would not emphasize too much on a causal discrimination story, and would just tell an interesting correlation between UV radiation and employment, and acknowledge it could due to discrimination or some other reasons.

Indeed, we cannot refute the claim about people’s tendency to search less when UV radiation is greater. However, we do not think this should be of a great concern thanks to the fact that the effects are significant only for people with intermediate skin tones. We do think this supports our discrimination hypothesis and so does the fact that effects are significant only for men. Finally, note that the effects of UV radiation on employment are found when the week and the year are held constant. 

2. The authors find that UV radiation only has effect on men’s employment but not women’s, and interpret it as “this suggests that it is discrimination against men that drives the results we observe” (page 16, last sentence). Again, one can think of alternative stories other than discrimination. Perhaps women are more likely to use sunscreen and less likely to get darker. Or perhaps women drop out of labor force (e.g., stay home and take care of children) rather than become unemployed. In the latter case, maybe the authors can provide analysis on labor force participation, in addition to employment.

Thank you for this comment. Following your suggestion, we include this additional alternative explanation to the manuscript. We note however that studies have shown that most adults do not use sunscreen on the face (page 18). 

3. Does NLSY surveys every year and the weekly employment status is based on recall? Recalling employment status every week for the past year may not be accurate and is subject to large measurement error. The authors should discuss that.

Following the comment, we discuss this in the ‘data and methods’ section of the paper. 

4. During Christmas season or winter break, many people travel to other city or state and the UV radiation of where they usually reside (I assume the place of reside only reports once each survey year in NLSY) would have no effect on their skin tone. Since the data provide interview dates, perhaps the authors can exclude holiday season where people travel the most, as a robustness check. People can be out of town in other weeks too, but it maybe difficult to detect that.

We tried running 4 different models (one for each season) testing the effects of UV radiation on employment: The UV radiation in the winter has no effect on the employment of participants (but the UV radiation in the fall, summer and spring does). this may be because: (a) winter break (as suggested). (b) the smaller sample size. (c) people don’t go out when it’s winter (4) people don’t tan (don’t look darker) when UV radiation is very low (like in the winter). We are not sure how to disentangle these 4 different mechanisms. 

Minor comments:

5. The authors conduct several solid regression analysis. I would suggest the authors to write out the regression equation explicitly before showing the results.

We added the equations to the paper. 

6. In addition to group the data into three skin tone categories (light, intermediate, dark), I would like to see results in finer categories and not to group them together (perhaps in the appendix), given the data have a large sample size.

The dataset is indeed large but has a relatively small number of unique individuals. Using the 1-10 skin tone categories generates very small sample sizes. Instead, we try to group them to different categories, for robustness tests (see the appendix, Table A4). 

7. Table A2 is not mentioned in the text, perhaps it should be referred to in the last paragraph of page 10? What’s the difference between column (1) and (2) in Table A2? Perhaps there is a missing “Y” in the column (2) for dummies, so does column (3)?

Yes, following the comment we updated the manuscript and table.

---

## [Decision Letter · Decision Letter 1]

16 Jun 2020

Those who tan and those who don’t: A natural experiment on Colorism

PONE-D-20-03529R1

Dear Dr. Kricheli-Katz,

Both reviewers are happy with your revision. Reviewer 1 has a careful comment but I think your revision has already addressed it. Therefore, we’re pleased to inform you that your manuscript has been judged scientifically suitable for publication and will be formally accepted for publication once it meets all outstanding technical requirements.

Kind regards,

Shihe Fu, Ph.D.

Academic Editor

PLOS ONE

Additional Editor Comments (optional):

Reviewers' comments:

Reviewer's Responses to Questions

**Comments to the Author**

1. If the authors have adequately addressed your comments raised in a previous round of review and you feel that this manuscript is now acceptable for publication, you may indicate that here to bypass the “Comments to the Author” section, enter your conflict of interest statement in the “Confidential to Editor” section, and submit your "Accept" recommendation.

Reviewer #1: (No Response)

Reviewer #2: All comments have been addressed

2. Is the manuscript technically sound, and do the data support the conclusions?

Reviewer #1: Yes

Reviewer #2: Yes

3. Has the statistical analysis been performed appropriately and rigorously? 

Reviewer #1: Yes

Reviewer #2: Yes

4. Have the authors made all data underlying the findings in their manuscript fully available?

Reviewer #1: Yes

Reviewer #2: Yes

5. Is the manuscript presented in an intelligible fashion and written in standard English?

Reviewer #1: Yes

Reviewer #2: Yes

6. Review Comments to the Author

Reviewer #1: Referee Report

PONE-D-20-03529R1

Those Who Tan and Those Who Don’t: A Natural Experiment on Colorism

First of all, I would like to apologize for the misunderstanding on results in Table 2 reported by the previous version of this paper. The authors report the odds ratio rather than the log odds ratio, which explains the negative effects of UV on employment with coefficients greater than 0 but smaller than 1.

The only main concern left is about the larger sample size of the intermediate skin tones. This relates to the important finding of the paper that the negative employment effect is only found statistically insignificant for people with intermediate skin tones, who are mostly prone to tan. The authors may want to dismiss the concern that the statistical significance is due to a larger sample. Indeed, authors mentioned that they have done the work and found similar results based on a randomly-picked subsample of the intermediate tones.

My previous report suggests the authors reporting the results based on these random subsamples (General Comments, bullet c). Perhaps I was not clear enough and the authors do not provide the results in the current version. Actually, a table similar to the Table 2 suffices. The only thing to be done is to show the results based on a random sample of intermediate skin tones with balanced sample size, such as the group of 583 (301231 observations) that the authors have done with. It is good that the authors have tried for different random subsamples of intermediate skin tones. I am not asking to report the results based on all these random subsamples. As they are generated randomly, results based on one of these subsample is enough to show the robustness of the results in Table 2.

Of course, the new Table 2 reports results based on the log odds ratio as well as the standard errors. From these new statistics, we can conveniently calculate the z-values and assess how standard errors matter for the significance. A sample size similar to the lightest tones or darkest tones is likely to produce the standard error similar to these two groups (i.e. 0.11 or 0.12). Under these standard errors, the estimated effects for intermediate tones (i.e. -0.027 and -0.041) will still be significant. Therefore my previous concern about the sample size can be partially addressed.

Reviewer #2: (No Response)

7. PLOS authors have the option to publish the peer review history of their article (what does this mean?). If published, this will include your full peer review and any attached files.

Reviewer #1: No

Reviewer #2: No

---

## [Editor Report · Acceptance letter]

23 Jun 2020

PONE-D-20-03529R1 

Those who tan and those who don’t: A natural experiment on Colorism 

Dear Dr. Kricheli Katz:

I'm pleased to inform you that your manuscript has been deemed suitable for publication in PLOS ONE. Congratulations! Your manuscript is now with our production department. 

Kind regards, 

on behalf of

Dr. Shihe Fu 

Academic Editor

PLOS ONE